# Overview of the Polyphenols in Salicornia: From Recovery to Health-Promoting Effect

**DOI:** 10.3390/molecules27227954

**Published:** 2022-11-17

**Authors:** Francesco Limongelli, Pasquale Crupi, Maria Lisa Clodoveo, Filomena Corbo, Marilena Muraglia

**Affiliations:** 1Dipartimento di Scienze del Suolo e Degli Alimenti, Università degli Studi di Bari, Campus Universitario E. Quagliarello Via Orabona 4, 70125 Bari, Italy; 2Dipartimento Interdisciplinare di Medicina, Università degli Studi Aldo Moro Bari, Piazza Giulio Cesare 11, 70124 Bari, Italy; 3Dipartimento di Farmacia-Scienze del Farmaco, Università degli Studi di Bari, Campus Universitario E. Quagliarello Via Orabona 4, 70125 Bari, Italy

**Keywords:** Salicornia, extraction technologies, phenolic acids, flavonoids, lignans

## Abstract

Nowadays, there has been considerable attention paid toward the recovery of waste plant matrices as possible sources of functional compounds with healthy properties. In this regard, we focus our attention on Salicornia, a halophyte plant that grows abundantly on the coasts of the Mediterranean area. Salicornia is used not only as a seasoned vegetable but also in traditional medicine for its beneficial effects in protecting against diseases such as obesity, diabetes, and cancer. In numerous research studies, Salicornia consumption has been highly suggested due to its high level of bioactive molecules, among which, polyphenols are prevalent. The antioxidant and antiradical activity of polyphenols makes Salicornia a functional food candidate with potential beneficial activities for human health. Therefore, this review provides specific and compiled information for optimizing and developing new extraction processes for the recovery of bioactive compounds from Salicornia; focusing particular attention on polyphenols and their health benefits.

## 1. Introduction

Recently, the use of natural substances as a health resource has become increasingly popular, especially the recovery of antioxidants, minerals, pigments, polymer, and oils from fresh vegetable matrices and agro-industrial by-products. Furthermore, nutraceutical extracts from food plants are finding success due to their nutritional and functional properties [1,2].

Salicornia is a halophyte plant that grows in saltwater around much of the Mediterranean coast as well as in the coastal regions of East Asia. The most-studied species of Salicornia, illustrated in Table 1, are the following:*S. europea*, the common glasswort, appears as a relatively small plant, having bright green stems characterized by small leaves and fleshy fruits that contain a single seed. It is present in Britain, France, and Ireland [3];*S. bigelovii*, the dwarf saltwort Salicornia, is located in USA and Mexico and can be distinguished from other species by its acute and sharply mucronate leaf and bract tips [4];*S. ramosissima*, also known as purple glasswort, is situated in France and Iberia and has stems up to 30 cm high, highly branched, green or purple depending on their youth [5];*S. herbacea*, the dwarf glasswort, is diffused in Korea and Italy and has fleshy, erect stems and opposite leaves, similar in appearance to flattened scales on the stems [6];*S. brachiata*, also named umari keerai, is located in India [7].

**Table 1 molecules-27-07954-t001:** Characteristics of the main Salicornia plants.

	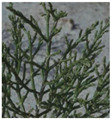	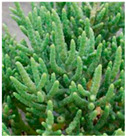	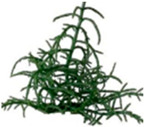	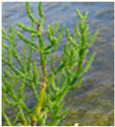	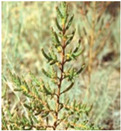
**Botanical name**	*S. bigelovii*	*S. brachiata*	*S. europea*	*S. ramosissima*	*S. herbacea*
**Common names**	Dwarf saltwort	Umari Keerai	Common glasswort	Purple glasswort	Dwarf glasswort
**Geographical range**	USA, Mexico	India	Britain, France, Ireland	France, Iberia	South Korea
**References**	[4]	[7]	[3]	[5]	[6]

Salicornia has long been used in traditional Eastern medicine to treat intestinal disorders (diarrhea and constipation) and inflammatory disorders, including nephropathies, hepatitis, diabetes, and cancer (Figure 1). Furthermore, the effects of Salicornia extract on pain were recently evaluated by performing a placebo-controlled study to investigate its analgesic and antipruritic effects, e.g., the effects on neurogenic inflammation at 24 and 48 h after the topical application of *S. ramossisma* in healthy subjects [8]. The collected results represent a starting point both for the evaluation of the antinociceptive potential of the bioactive compounds of Salicornia and for overcoming the limits derived from its pharmaceutical synthesis in this sector [9].

It is acknowledged that saline stress leads to the production of reactive oxygen species (ROS) that damage cell membranes and enzymatic activity in plants [10]. Salicornia is one of the most salt-tolerant species and is, therefore, equipped with a powerful antioxidant system (including enzymes and antioxidant compounds) capable of extinguishing and resisting ROS, and is also responsible for its therapeutic effect against the aforementioned chronic diseases. Therefore, the medicinal qualities of this plant have aroused the interest of the scientific community, spurring investigation of its phytochemical composition. Phytochemicals found in Salicornia, such as tungtunmadic acid, quercetin, chlorogenic acid, and their glycosides, contain many hydroxyl groups that make them highly electrophilic; this feature is fundamental because it induces the stimulation of antioxidant enzymes and exerts a radical scavenging activity, thus protecting cells from the damage provoked by ROS [11].

Nowadays, there is a growing interest in the valorization of such food plants and their eventual by-products for the recovery and/or biotransformation in various industrial applications, such as food supplements, cosmetics, and the pharmaceutical industry [12]. For this reason, more cost-effective and sustainable extraction methods to obtain nutraceutical extracts have been developed in the last few years [13,14]. Different extraction techniques have been used to isolate phytochemicals in Salicornia extracts, ranging from the more traditional (such as solid–liquid extraction by maceration) to non-conventional emerging technologies (such as microwave and ultrasound-assisted extraction and supercritical fluid extraction) in order to favor mass transfer, shorten extraction times, and/or reduce solvent requirements. Moreover, the technological parameters (i.e., particle size, solvent type and composition, solid-to-solvent ratio, extraction temperature and pressure, extraction time, and pH) affecting the extractability of bioactive compounds have been carefully optimized [15].

This review deals with an overview of the principal phytochemicals (especially polyphenols) identified in Salicornia that have shown health benefits, as well as the most widespread extraction methods (including green processes) employed for their recovery.

## 2. Principal Phytochemicals (Especially Polyphenols) Identified in Salicornia

Several studies on the characterization of Salicornia extracts by analytical techniques (i.e., HPLC-UV, HPLC-MS, GC-MS, etc.) have reported the presence of vitamins, amino acids, minerals, sterols, fatty acids, saponins, oxalates, phenolic acids, flavonoids, and lignans [16,17].

Salicornia has shown high levels of vitamin C; the content of ascorbic and dehydroascorbic acids is more than 100 mg/100 g. *S. bigelovii* shows a high amount of β-carotene (15.9 mg/100 g of fresh weight (fw)), which is a good source of vitamin A [10,18]. Lima et al., 2020 have reported a high content of the soluble fat vitamin E, α-Tocopherol (241 μg/100 g fw), and vitamin B_2_ and B_5_ (with a relative content of 4.14% in 0.5 g) in *S. ramosissima* and *S. bigelovi*, respectively [19,20].

Significant contents of important amino acids, such as aspartic acid (140.1 and 165.5 mg/100 g fw), glutamic acid (160.5 and 182.3 mg/100 g fw), and isoleucine (107.5 and 94.7 mg/100 g fw) were quantified in stems and roots of *S. herbacea*. While, six amino acids, including glutamic and aspartic acids, which were present at the highest concentration, were found in *S. bigelovii* [16,18].

Na^+^ and K^+^ were the most abundant minerals in the Salicornia species; indeed, especially *S. bigelovi* and *S. europea* could be used as a salt substitute due to their protective effect on vascular dysfunction and hypertension. Recent studies proved that a high intake of salt increased blood pressure in rats, while intake of Salicornia, with the same quantity of sodium, had a low effect on blood pressure [21,22]. Up to 1421.2 and 173.3 mg/100 g of Na^+^ and K^+^, respectively, were found in *S. europea* leaves. In addition, important amounts of other minerals were also identified mainly in stems and roots of *S. herbacea* and *S. europea,* such as Ca^2+^ (158.8 mg/100 g), K^+^ (740.1 mg/100 g), Mg^2+^ (52.2 mg/100 g), and Fe^2+^ (3.25 mg/100 g) [23,24]. Moreover, the presence of selenium, an essential micronutrient with a significant antioxidant effect, was found in *S. brachiate* [25].

Sterols have been identified in *S. herbacea* and *S. europea* extracts. The main sterols contained are stigmasterol and ergosterol. Stigmasterol (0.47 mg/kg) shows hypoglycemic, antioxidant, antitumor, antimutagenic, and anti-inflammatory properties while ergosterol (0.31 mg/kg) is an important source of vitamin D and possesses potent NF-kB inhibitory activities [11,26]. Another sterol identified is the ß-sitosterol, the predominant phytosterol in the human diet, with neuroprotective and anti-diabetic effects [27,28].

Salicornia shows also a relevant amount of saturated and polyunsaturated fatty acids, including stearic acid, linolenic acid, linoleic acid, and palmitic acid, especially found in *S. herbacea*, *S. europea*, and *S. brachiata* species [29,30]. Linolenic acid is the most representative polyunsaturated fatty acid mainly found in Salicornia leaves at concentrations around 4 mg/g [31]. This compound is used in the prevention and treatment of many common diseases such as arthritis, eczema, and premenstrual syndrome [31,32]. Linoleic acid, mostly contained in leaves (2.18 mg/g), is an essential fatty acid whose consumption is fundamental to good health. It exhibited high antioxidant and antiproliferative activities toward HepG2 and A549 cells, as shown by Wang et al., 2013 [11].

Saponins were identified in S. *herbacea* and *bigelovii* extracts with antioxidant and antifungal effects. Among these, new noroleanane and nortriterpene saponins were isolated. Nevertheless, saponins could exhibit toxicity due to tissue necrotic and gut permeability alteration, which can compromise the immune system [30]. Similarly, oxalates, which were revealed in high content, are anti-nutrients that should be removed to justify usage of Salicornia as a ‘sea vegetable’ [30].

Other identified aliphatic compounds are octacosanol and tetracosanol, two aliphatic alcohols isolated from *S. europea* and *ramosissima*. Tetracosanol has a significant role in diabetes therapy for its α-amylase ability, while Octacosanol has been known for its ability to lower total cholesterol levels [30,33].

However, despite the significant amount of the aforementioned nutrients present in Salicornia tissues, major attention has been devoted to their phenolics content, which ranges from 1.2 to 2 mg GAE/g fw [34], due to the related health benefits and bioactivity [35]. Phenolic compounds, characterized by aromatic rings with one or more hydroxyl groups, include several structurally different classes; those found in Salicornia extracts at different concentration levels are mainly phenolic acids, flavonoids, and lignans [28] (Figure 2).

### 2.1. Phenolic Acids

Phenolic acids are a class of polyphenols very abundant in the human diet, as they are contained in vegetables, fruits, and whole grains. The daily intake of phenolic acids should be around 200 mg/day [36]. Phenolic acids are types of aromatic acid compounds derivatives of benzoic and cinnamic acids. They exhibit high antioxidant activity and thus possess health protective effects, such as antimicrobial, anticancer, anti-inflammatory, and anti-mutagenic. [37]

Salicornia has been shown to possess a high content of phenolic acids, including chlorogenic, tungtungmadic, ferulic, protocatechuic, caffeic, salicylic, syringic, and coumaric acids [23,38].

Chlorogenic acid, an ester of caffeic acid, is found in many plants, including Salicornia, with a content of 0.84 mg/g of dry weight (dw) from ethanolic extract of *S. europea* [39]. Furthermore, various chlorogenic acid derivatives, such as dicaffeoylquinic acids, have been also identified in Salicornia. For instance, four new dicaffeoylquinic acid derivatives, specifically 3-caffeoyl-5-dihydrocaffeoylquinic acid (75.6 ± 2.3 mg/100 g fw), 3-caffeoyl-5-dihydrocaffeoylquinic acid methyl ester (69.3 ± 1.4 μg/100 g fw), 3-caffeoyl-4-dihydrocaffeoylquinic acid methyl ester (71.9 ± 1.9 μg/100 g fw), and 3,5-dihydrocaffeoylquinic acid methyl ester (171.9 ± 1.5 μg/100 g fw), as well as 3-caffeoylquinic acid and 3-caffeoylquinic acid methyl ester, were isolated from the aerial parts of *S. herbacea*. The antioxidant activity of these compounds was examined by measuring the cholesteryl ester hydroperoxide (CE-OOH) content, produced by oxidation in healthy human plasma, during rat blood plasma oxidation induced by copper ions. CE-OOH accumulates in atherosclerotic plaques, and for this reason, it has been used as an index of lipid peroxidation to evaluate the inhibitory effect of antioxidants on lipids oxidized in blood plasma. The caffeoylquinic acid derivatives considerably inhibited CE-OOH. In particular, the dicaffeoylquinic acid derivatives showed a relatively higher ability to inhibit CE-OOH formation rather than monocaffeoylquinic acid derivatives and caffeic acid. Moreover, this important activity is demonstrated by their significant scavenging effects, as shown by the DPPH assay performed. DPPH radical-scavenging activities and metal-chelating effects of caffeoylquinic acid derivatives were proportionally correlated with the number of catechol groups. Therefore, the dicaffeoylquinic acid derivatives, isolated from *S. herbacea*, containing a catechol group may act as excellent radical scavengers and metal-chelating agents [40].

3-caffeoyl-4-dicaffeoylquinic acid (CDCQ), known as tungtungmadic acid, has been isolated from *S. herbacea*. It is a natural chlorogenic acid derivative with high antioxidative activity in free radical scavenging assay and in the iron-induced liver microsomal lipid peroxidation test. In addition, it shows a significant effect in protecting the plasmid DNA against strand breakage induced by Fe^3+^-nitrilotriacetic acid-hydrogen peroxide [38].

CDCQ antioxidant, anti-inflammatory, and anticancer effects were demonstrated on nitric oxide (NO) production and eNOS phosphorylation in endothelial cells. CDCQ induced eNOS phosphorylation and NO production by the phosphorylation of PKA, CaMKII, CaMKKβ, and AMPK in endothelial cells. Furthermore, CDCQ exhibited activities on the L-type Ca^2+^ channel (LTCC), present in the membrane of endothelial cells, and mediated signaling by increasing intracellular Ca^2+^ influx (Figure 3) [41]. Overall, CDCQ exercises a significant protective effect on endothelial cell function, which is important for modulating hemostasis, blood flow, and vascular health [42,43].

Ferulic acid is abundant in plants and it is mainly conjugated with mono- and oligosaccharides and lipids. *trans*-ferulic acid (TFA) was the most abundant component identified in desalted *S. europea* powder (DSP), exhibiting significant biological effects. When administrated to rats with HFD-induced obesity, DSP reduced body weight gain, abdominal fat mass, and serum lipid profiles. Furthermore, it inhibited adipogenesis and adipocyte differentiation by down-regulating the adipocyte-specific transcriptional regulators SREBP1, FAS, C/EBPa, and PPARc. Therefore, these results show that DSP and, in particular, TFA could protect against HFD-induced obesity [44].

Another research examined the biological effects of *trans*-ferulic (2.60 ± 0.33 mg/g) and *p*-coumaric (3.19 ±0.47 mg/g) acids isolated from *S. europea*. Notably, the two acids contained in the Salicornia extracts were investigated for their effect on vascular dysfunction and hypertension. *p*-coumaric acid and *trans*-ferulic acid-induced vasorelaxation in a dose-dependent manner in the coronary artery on vascular dysfunction induced by high salt content [21].

Wang et al., 2013, isolated (from *S. herbacea*) for the first time the pentadecylferulate, an alkyl ester of ferulic acid. This compound showed higher antioxidant activity than ascorbic acid and other ferulic acid derivatives in the DPPH assay, probably due to its structure characterized by a long-chain alkane (15-carbon) moiety. In addition, it showed anticancer activity in cells of hepatocarcinoma HepG2 and pulmonary adenocarcinoma A549 [11].

Various extracts of *S. europaea* collected from two different regions of Tunisia were evaluated for their antimicrobial activity, using the agar well diffusion method. The two different ethanolic extracts exhibited a high total phenolic content (TPC, 43.1 ± 0.2 and 32.1 ± 0.3 mg GAE/g dw) with a rich composition in phenolic acids, including gallic acid (0.8 mg/g dw), chlorogenic acid (0.22 mg/g dw), vanillic acid (0.69 mg/g dw), caffeic acid (0.28 mg/g dw), and coumaric acid (0.32 mg/g dw), but also in catechin hydrate (1.17 mg/g dw) and rutin hydrate (10.05 mg/g dw). Because of the structural difference in their lipid layer, Gram-positive bacteria were significantly more sensitive to the two extracts and showed more inhibition zones compared to Gram-negative bacteria [24]. Thus, a synergistic or additive effect of the phenolic acids with the other compounds could be hypothesized for the antimicrobial activity of *S. europaea* extracts [24,45]. Another study [46] confirmed the antimicrobial activity of other halophyte plants, such as *S. brachiate*, whose extracts were also very efficient against Gram-positive bacteria.

The effect of *S. ramosissima* ethanolic extract was evaluated on carbon tetrachloride (CCl_4_)-induced testicular damage in a mouse model. The phytochemical composition of the extract showed the presence of known phenolic and aliphatic compounds, such as ethyl linolenoate (0.44 mg/g dw), sitostanol (0.09 mg/g dw), octadecyl (0.12 mg/g dw) and eicosanyl (0.09 mg/g dw) (E)-ferulates, ethyl (E)-2-hydroxycinnamate (0.06 mg/g dw), and scopoletin (0.09 mg/g dw). The results of the histopathological analysis showed that the treatment with the ethanolic extract prior to CCl_4_ administration significantly prevented the architectural disorder of seminiferous epithelium and germ cell exfoliation. Therefore, the nature of the extracted compounds suggests a significant therapeutic value on the male reproductive system, especially due to the antioxidant action of its constituents, besides other therapeutic and nutritional effects of certain compounds isolated [47].

### 2.2. Flavonoids

Flavonoids are the most important phytochemical compounds present in many plants, fruits, and vegetables, which confer many health benefits, including anticancer, antioxidant, anti-inflammatory, antiviral, neuroprotective, and cardioprotective effects [48]. Their chemical structure, in particular the presence of hydroxy groups, influences their bioavailability and biological activity [49,50]. Flavonoids possess a basic 15-carbon flavone skeleton with two benzene (A-B) rings connected by a three-carbon pyran ring. The position of the catechol B-ring and the number of hydroxy groups on the B-ring influence their antioxidant capacity [51,52].

Rutin, isorhamnetin, quercetin, quercetin 3-O-β-D-glucopyranoside, isorhamnetin 3- O-β-D-glucopyranoside, noreugenin, and isoquecitrin are the most prevalent compounds in Salicornia. These compounds, especially quercetin, rutin, and isorhamnetin 3-O-β-D-glucopyranoside, have relevant anti-diabetic, cardiovascular protection, and anticancer activity due to their antioxidant potential [53,54]. Isorhamnetin 3-O-β-D-glucopyranoside, isolated from *S. herbacea*, exhibited scavenging intracellular radical activity in the cellular system, as well as in the cell-free system. Moreover, it participated in the modulation of cellular redox status due to the induction of cellular GSH levels and other antioxidant enzymes. Therefore, this compound showed a relevant effect in the prevention of radical-mediated cellular damage and could be developed as a candidate for potential use as a natural antioxidant related to oxidative stress [55]. Significant content (up to 57.52% of total compounds) of flavonoids and flavonoid glycosides have been revealed in *S. bigelovii* and *S. europaea* extracts. Other studies reported in the literature have isolated rutin hydrate (10.05 mg/g dw) and catechin hydrate (1.17 mg/g dw) from *S. europea* [24,39,56].

Kim et al., 2011 isolated a novel flavonoid glycoside, isoquercitrin 6”-O-methyloxalate from *S. herbacea*; this compound, along with quercetin 3-O-β-D-glucopyranoside, exhibited a relevant antioxidant activity compared to other flavonoids compounds that presented substitutions on catechol group of B ring [57]. Therefore, the catechol group of the B ring has an important effect on the pharmaceutical property of the molecule. In particular, it influences the antioxidant activity of flavonoids as reported by previous studies [58].

The flavonol glycosides, isoquercitrin 6″-O-methyloxalate, isorhamnetin 3-O-β-D-glucopyranoside, and quercetin 3-O-β-D-glucopyranoside from *S. herbacea* extract have been examined in macrophages and trophoblasts to test their inflammatory properties. Pretreatment and delayed treatment of these compounds in bone marrow-derived macrophages (BMDMs) reduced the activity of NLRP3 inflammasome induced by lipopolysaccharide (LPS) and adenosine triphosphate stimulation and downregulated interleukin (IL)-1β production. In addition, the extract decreased the mRNA expression of NLRP3, IL-1β, and IL-6 in the LPS-stimulated human trophoblast cell line inhibiting the production of IL-6 and IL-1β and decreasing the expression of cyclooxygenase-2 [59].

Other interesting flavonoids have been isolated from *S. Europea* such as luteolin (0.019 mg/g dw), kaempferol (108.1–247.6 mg/100 g dw) [60], and its glucoside derivative [29]. Like previous molecules, these compounds also have antioxidant, anti-inflammatory, antiapoptotic, and cardioprotective activities [61].

Another study explored the effects of the ethyl acetate fraction of desalted *S. europea* extract on atherosclerotic events in vascular smooth muscle cells (VSMCs) and vascular neointima formation. The major abundant components were five phenolic acids and four flavonols, including protocatechuic acid (8.4 mg/g), chlorogenic (14.1 mg/g), caffeic acid (9.5 mg/g), *p*-coumaric acid (6.8 mg/g), ferulic acid (8.2 mg/g), quercetin-3O-b-D-glucopyranoside (3.4 mg/g), isorhamnetin-3O-b-D-glucopyranoside (16.2 mg/g), quercetin (2.5 mg/g), and isorhamnetin (18.4 mg/g). The study demonstrated that the extract suppressed the platelet-derived growth factor (PDGF)-BB-induced migration and proliferation of VSMCs and increase mitogen-activated protein kinases (MAPK) and extracellular signal-regulated kinase (ERK)1/2 phosphorylation in VSMCs, consequently leading to the reduction of neointimal hyperplasia (Figure 4) [62].

Essaidi et al., 2013 highlighted how *S. herbacea* methanol extract was a potent inhibitor of cytochrome P450 CYP1A2, CYP3A4, and CYP2D6, related to the activation of carcinogens, due to its phenolic composition characterized by the presence of many phenolic acids and flavonoids, such as chlorogenic acid, caffeic acid, syringic acid, *p*-coumaric acid, sinapic acid, ferulic acid, salicylic acid, myricetin, quercetin, and kampferol [63,64]. It is demonstrated that flavonols and flavonoids, in general, such as quercetin and kaempferol have a 3-OH and phenolic hydroxyl (eOH) group with inhibitory activity on CYP2D6, CYP1A2, and CYP3A4 [65].

### 2.3. Lignans

Lignans are a class of natural compounds widely produced by various plant species. Recent studies have demonstrated that lignans provide very important biological and pharmacological properties, including immunosuppressive, anti-inflammatory, cardiovascular, antioxidant, antitumor, and antiviral effects. In fact, they are found in many plants used in Eastern medicine [66,67]. Lignans contain a core scaffold that is formed by two or more phenylpropanoid units and monomers such as cinnamic acid, cinnamyl alcohol, propenyl benzene, and allyl benzene [68].

Syringaresinol 4-O-β-D-glucopyranoside, erythro-1-(4-O-β-D-glucopyranosyl-3,5-dimethoxyphenyl)-2-syringaresinoxyl-propane-1,3-diol, and longifloroside B have been the main lignans isolated from *S. europea*. Notably, syringaresinol 4-O-β-D-glucopyranoside has shown interesting biological activities including DPPH radical scavenging activity, antiestrogenic activity against MCF-7 cells, and antitumor activity against the A549 cancer cell line [69]. This compound has been found to modulate lipid and glucose metabolism in HepG2 cells and C2C12 myotubes [70]. In addition, other lignans such as (-)-syringaresinol and episyringaresinol-4″O-β-D-glucopyranoside were isolated from *S. europaea*. In particular, (-)-syringaresinol (4.25 mg/Kg dw) has been found to possess antiplatelet aggregation, DPPH radical scavenging, nitric oxide (NO) inhibition, and P-glycoprotein inhibition activities [28].

Acanthoside B is another important lignan, extracted from S. europaea, which has exhibited negligible toxicity and showed a dose-dependent nitric oxide inhibitory potential in Lipopolysaccharide (LPS)-stimulated BV-2 microglial cells. Furthermore, it has attenuated scopolamine-inflicted AD-like amnesic traits by restoring the cholinergic activity, suppressing neuroinflammation, and activating the TrkB/CREB/BDNF pathway in mice. For these effects, acanthoside B could be a potential drug candidate for the treatment of neurodegenerative diseases (Figure 5) [71].

## 3. Comparison of Extraction Methods of Polyphenols from Salicornia

Extraction from plants is generally the crucial step for both the isolation and exploitation of their bioactive compounds [72]. The bioactive compounds can be extracted from fresh or dried plant samples firstly pre-treated by milling, grinding, and homogenization. Furthermore, the fundamental parameters affecting the extraction yield, such as extraction time and temperature, solvent-to-solid ratio, number of repetitions, as well as the choice of extraction solvents, can be opportunely optimized [13,73]. Ideally, an extraction method should be quantitative, non-destructive, and time-saving; moreover, it should be chosen based on the chemical composition and purity degree of the extract one wishes to obtain [74].

Conventional solid-liquid extraction by maceration in organic solvents has been traditionally used to extract phenolic compounds from vegetable matrices. The main disadvantages of this process include the length of time needed in solid–solvent contact to reach equilibrium and the overall use of high temperatures which, although can favor the solid-liquid mass transfer of the compounds, frequently provokes a decrease in the extraction efficiency due to thermosensitivity of polyphenols [75]. As a consequence, in order to solve these limitations, new techniques have been developed in recent years for the extraction of bioactive compounds from plant materials, including ultrasound-assisted extraction, microwave-assisted extraction, enzyme-assisted extraction, and supercritical fluid extraction [76].

### 3.1. Maceration

Maceration is a traditional method, based on the soaking of plant materials in a solvent for the recovery of bioactive compounds. It uses organic solvents, such as methanol, acetone, and ethyl acetate, which are toxic to human health and the environment because some residues may remain in the final extract. Moreover, this method is associated with high solvents volumes and extraction times that generate a large amount of waste and low extraction yields. Another disadvantage is the high temperature that sometimes leads to the loss of polyphenols caused by hydrolysis and/or oxidation during the process [77,78].

There are many reports on the investigation of the parameters affecting the profile and yield of polyphenols from Salicornia by maceration. Extraction with the sample added to boiling distilled water for 5 min has shown a phenolic composition of *S. ramosissima* extracts that are mainly characterized by the presence of myricetin, gallic acid, catechin, rutin, kaempferol-3-O-glucoside, and quercetin-3-O-galactoside [79].

Other studies have performed maceration at room temperature with a longer extraction time (16–24 h) [58,80]. This technique ensured the recovery of many phenolic compounds that could be degraded at high temperatures. Extraction with methanol from aerial parts of *S. herbacea* has identified four new dicaffeoylquinic acid derivatives and some flavonoid glucosides [40].

Modulating the polarity of the extraction solvents allows for the obtainment of a different type/range of polyphenols. High amounts of phenolic compounds are mainly found in extractions with a polar solvent. Ethanol extracts of *S. europea* have shown the presence of nine polyphenols including phenolic acids (i.e., chlorogenic acid or gallic acid) and flavonoids (i.e., rutin hydrate and catechin hydrate) [24]. While *S. herbacea* extraction with 80% acetone at room temperature for 24 h has allowed recovering pentadecyl ferulate in high yield [11].

Five phenolic acids, including protocatechuic acid, chlorogenic acid, caffeic acid, *p*-coumaric acid, and ferulic acid, together with quercetin-3-O-β-D-glucopyranoside, isorhamnetin-3-O-β-D-glucopyranoside, quercetin, and isorhamnetin, were identified in *S. europea* extracts obtained by a vacuum extractor at 100 °C for 5 h and subjected to ethanol precipitation for removing the high molecular polysaccharides and proteins [63].

Ferreira et al., (2018) executed extraction with fresh aerial parts (1.2 kg) of *S. ramosissima*, chopped into small pieces, and extracted with ethanol (5 L) three times at room temperature for 24 h using an overhead stirrer [47].

Another study performed an extraction on *S. ramosissima* biowaste using a thermostatic water bath. The powdered samples (2.5 g) were mixed with distilled water (25 mL) and extracted at different times and temperatures with a constant agitation of 200 rpm. The best extraction conditions were fixed at 80 °C in 10 min. Hydroxycinnamic acids, flavonols, and isoflavones, known for their excellent scavenging ability against ROS, have been extracted and isolated from *S. ramosissima* by-product which could be considered a useful source of antioxidant and neuroprotective compounds with interesting applications in the food, nutraceutical, and cosmetic industries [81].

### 3.2. Microwave-Assisted Extraction

Microwave-assisted extraction (MAE) is one of the most advanced methods currently used in plant extraction and is based on the effect of microwaves with a frequency between 0.3 to 300 GHz. Depending on the polarity of the solvent and the presence of ions in the solvent (methanol, ethanol, water, or their mixtures, having high or medium absorbance capacity of microwaves, are generally used), dielectric heating and ionic conduction can occur simultaneously in causing the swelling and rupturing the plant cell, thus facilitating the extraction of polyphenols [82].

The advantages of this method are mainly quick heating and extraction efficiency, lower solvent requirements, short extraction time, and a clean process. On the other hand, MAE is usually performed at higher temperatures (>80 °C), thus its application in the isolation of antioxidants has to be carefully assessed [83].

The only report dealing with the application of this technology to the extraction of polyphenols from Salicornia was that of Silva et al. (2021) who performed a MAE achieving better antioxidant and antiradical activities (65.56 mol FSE/g dw and 17.74 g AAE/g dw for FRAP and ABTS assays, respectively) rather than conventional extraction [79].

### 3.3. Ultrasound-Assisted Extraction

Ultrasound-assisted extraction (UAE) uses ultrasounds that enhance the extraction rate by increasing the mass transfer and possible rupture of cell walls due to the well-known “cavitation effect”, leading to higher product yields without modifying the extract composition [82].

Moreover, UAE allows for a reduction of processing time, thermal degradation losses, solvent, and energy consumption, and is compatible with any solvent including water and other generally recognized as safe (GRAS) solvents [84].

The influence of extraction time, temperature, solvent concentration, solid-to-liquid ratio, particle size, ultrasound power, and frequency on the polyphenols recovery from Salicornia were investigated in many studies.

Several researchers have performed extractions at 50 °C for different time periods in order to select the most suitable extraction time to recover the highest level of phenolic substances [46]. Even though an extraction time of 20 min has been shown as the best condition to recover quantities of antioxidant compounds from Salicornia matrix [85]. Regarding solvent type and concentration, different preferences appear in the literature, from 40% to 80% ethanol, for the best recovery of phenolic acids, phenolic alcohols, and flavonoids. The presence of water reduces solution viscosity and increases the plant swelling promoting a higher yield [86,87]. The effect of extraction time (1, 9, 30, 51, and 60 min) and power of ultrasonic waves (100, 150, 275, 400, and 450 W) parameters on the extraction of polyphenols from Salicornia were also investigated by recent reports. The highest content of phenolic compounds, ranging between 16.3 and 20.0 mg GAE/g dw, and overall antioxidant activity were obtained by extracting *S. ambigua* or *S. neei* samples (250 mg) with 15 mL of ethanol 80% in an ultrasound bath setting the ultrasound power and extraction time at 275 W and 30 min, respectively. Conversely, a longer extraction time led to lower contents of polyphenols, indeed lower extraction yield of gallic acid was found with an extraction time of 50 min [88,89].

Fifteen hydroxycinnamic acids (i.e., neochlorogenic, chlorogenic, *p*-coumaric, ferulic, and caffeic glycosides acids) together with caffeoylquinic dimers, such as 3,5-dicaffeoylquinic and 4,5-dicaffeoylquinic acids, and nine flavonoids (quercetin glycosides and apigenin glycoside) were extracted from *S. ramosissima* by adding 100 mL of ethanol:water (80:20, *v/v*) solution to 10 g of the fresh plant in an ultrasonic water bath at 40 kHz and 220 W for 60 min at 25 ± 3 °C [90].

Kim et al. 2009 performed the extraction from *S. herbacea* powder (5 g) by sonication but by changing the solvent composition and the extraction temperature with a constant working frequency of 40 kHz. A linear gain of total phenolic content with increasing ethanol concentration and decreasing extraction temperature was observed; indeed, the maximum phenolic content (50.36 mg GAE/g) was recovered at an ethanol concentration of 76.82% and extraction temperature of 50 °C, while the lowest phenolic content (33.09 mg GAE/g) was revealed at an ethanol concentration of 50% and extraction temperature of 40 °C [91].

Finally, Wang et al. recently applied an efficient association/combination of extractive methods. *S. bigelovii* samples were extracted by maceration using 16 mL of 60% ethanol for 3 h at 60 °C, followed by a 200 W ultrasonic extraction for 40 min. This technique has provided a high recovery of flavonoids, including rutin (20.23%), noreugenin (15.26%), isoquercitrin (7.13%), quercetin (2.54%), camellianin A (5.91%), and 7-O-β-D-glucopyranosyl-methoxychromone (1.04%) [20].

### 3.4. Supercritical Fluid Extraction

Supercritical Fluid Extraction (SFE) is another fast and efficient green method usually adopted for the extraction of natural compounds from plants. SFE employs supercritical fluids having low viscosity and high diffusivity, which, at the critical point (a specific temperature and pressure), diffuse into the solid matrix like gas and dissolve active materials like a liquid. These characteristics enhance diffusion and mass transfer while reducing extraction time [92]. The most widespread supercritical fluid in plant extractions is CO_2_, which has a very low critical temperature (31 °C) and can be easily removed, allowing rapid and selective extraction [93]. However, due to the apolarity of CO_2_, a co-solvent, such as water, ethanol, and methanol, is frequently used to allow the extraction of more polar compounds [94].

In this context, some researchers, carrying out the extraction of *S. europea* at constant extraction temperature (50 °C) and pressure (300 bar) with CO_2_ flow of 2 L min^−1^ and different ethanol concentrations (10, 20 or 40% *v*/*v*, respectively), have observed that both total phenolic compound (TPC), total extract yield (TEY), and antioxidant indices (ABTS and FRAP) were directly influenced by the ethanol percentage used in the SFE technique. Indeed, the values of TPC (0.29 vs. 0.15 mg GAEs g^−1^ dw), TEY (16.30 vs. 11.01 mg g^−1^ dw), ABTS (1.16 vs. 0.71 mg Tes g^−1^ dw), and FRAP (10.22 vs. 5.22 µmol FeSO_4_ g^−1^ dw) at 20% of ethanol were almost double than those at 10% [82,95]. The number of cycles (one cycle included 10 min of static condition and 10 min of dynamic phase) operated at the previous condition was crucial for yield: eight cycles were the optimal extraction condition to obtain a rich chemical profile [87].

Other studies use SFE for oil extraction from the matrix. This method allows for the extraction and analysis of the fatty acid profile of Salicornia oil. As shown by Adewale Folayan et al. (2019), high amounts of saturated and unsaturated fatty acids were obtained, such as myristic acid, palmitic acid, linoleic, and linolenic acid [96].

### 3.5. Enzyme-Assisted Extraction

Enzyme-assisted extraction (EAE) is a green extraction technology determined by numerous factors, including pH, temperature, treatment time, and enzyme selection. In particular, pH and temperature monitoring are essential to ensure optimal enzyme activity. The choice of the right enzyme allows the extraction of specific compounds from the matrix. For example, pectinase improved the extraction of polyphenols and phenols that are difficult to extract due to their high/low molecular weight [97]. Pectinase hydrolyses pectin macromolecules that constitute the cell membrane, exposing entrapped polyphenols to the solvent. Other types of enzymes generally used in the extraction are cellulase and hemicellulase, which participate in breaking down the cell wall [98]. EAE of Salicornia is often combined with other extraction techniques, as described in various studies. Indeed, extracts with high phenolics content (particularly, procathecuic, caffeic, and ferulic acids, and quercetin and isorhamnetin) and significant radical scavenging activity was obtained by coupling EAE and UAE through two steps: firstly, Salicornia was suspended in ionic water and treated with vyscozime (i.e., cellulotyc enzyme mixture) for 12 h at 40 °C; subsequently, it was extracted by ultrasound with ethanol for 3 h [99]. Isorhamnetin and acanthoside B were extracted from *S. europea* by treating the matrix with pectinase and cellulase at 50 °C for 15 h and subsequently by refluxing twice with 50% ethanol for 3 h [100].

Definitely, the main polyphenols present in the different types of Salicornia, together with their quantities, their extraction technologies, and the health benefits aforementioned, are summarized in Table 2.

## 4. Conclusions

Nowadays, evidence supporting the health effects of natural resources has led the scientific community to better understand the phytochemical composition of innovative health food diet products with the aim of obtaining bioactive compounds capable of reducing oxidative stress and related inflammatory disorders [101]. To this end, scientific evidence supports the beneficial effects of Salicornia, a halophytic plant of the Mediterranean basin and the coastal regions of East Asia, on gastrointestinal disorders, diabetes, hypertension, inflammation, vascular diseases, and oxidative stress. Focusing our attention on *S. herbacea*, *S. ramosissima,* and *S. europea*, the scientific studies collected in this review highlight the interesting content of phenolic compounds from 1.2 to 2 mg/GAE in Salicornia. Most of the research reviewed here supports the contribution of Salicornia in protecting cells from ROS-induced damage. Beneficial effects have been described, such as antitumor, antihypertensive, antibacterial, neuroprotective, and antidiabetic activities. In addition, a significant improvement in vascular disease was observed. Furthermore, in this review, a knowledge base for the selection of extraction procedures adopted for the recovery of polyphenols from Salicornia has been collected. The knowledge that emerged on the extraction methods adopted to extract polyphenols from Salicornia suggests the development and implementation of eco-friendly procedures to enhance the extraction of polyphenols through the adoption of sustainable extraction methods with higher yields.

Overall, the present work provides strong evidence that Salicornia polyphenols are involved in several pathways that contribute to both antioxidant and antiradical activities by candidating Salicornia for the development of nutraceuticals and food supplements with a wide variety of health-beneficial effects.

## Figures and Tables

**Figure 1 molecules-27-07954-f001:**
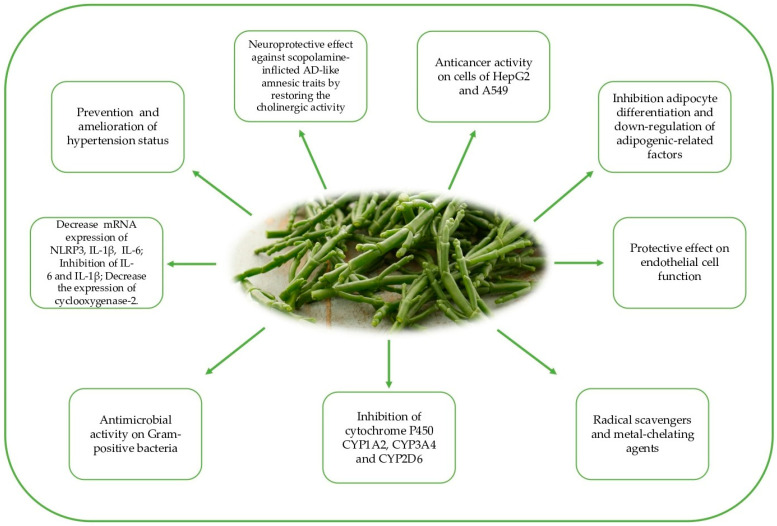
Health biological effects of Salicornia.

**Figure 2 molecules-27-07954-f002:**
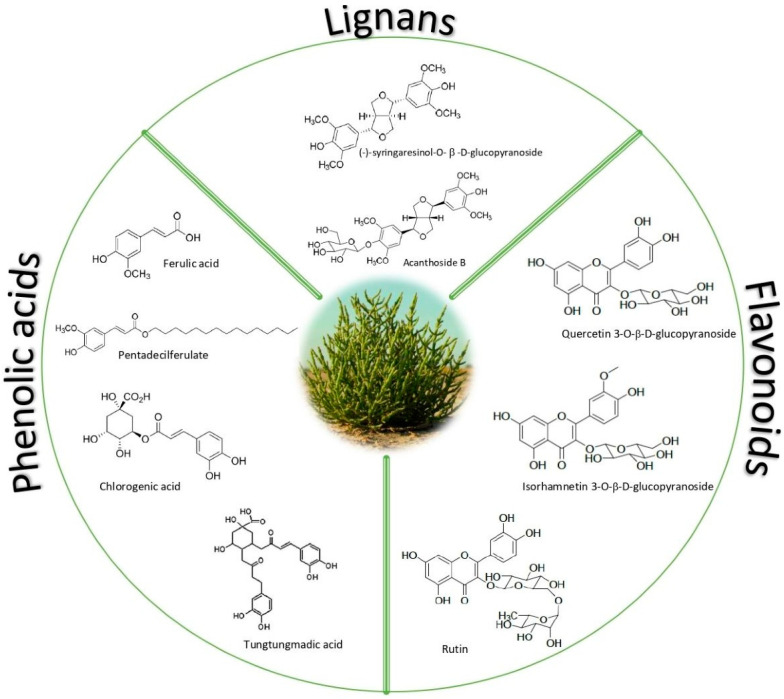
Classes of polyphenols revealed in Salicornia.

**Figure 3 molecules-27-07954-f003:**
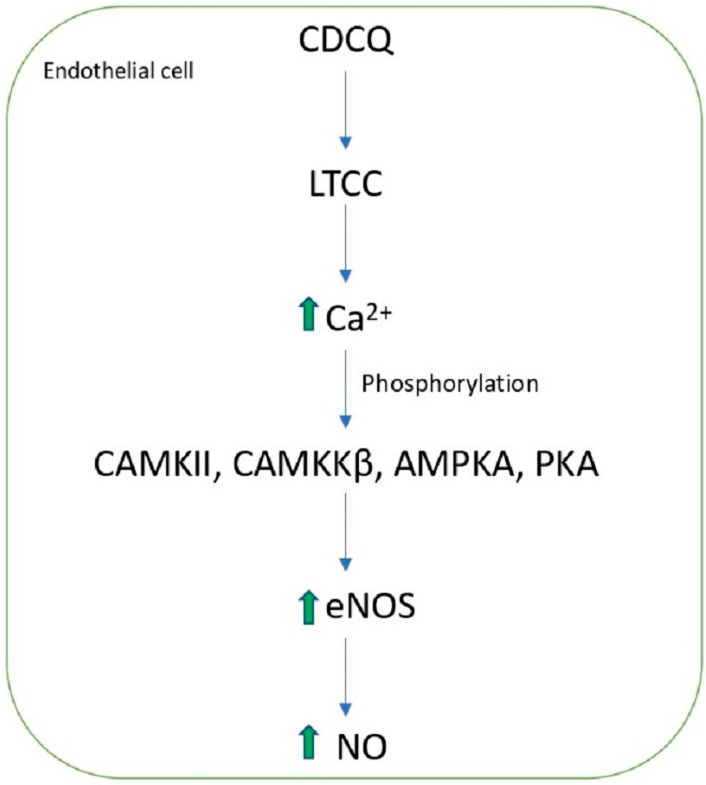
3-caffeoyl-4-dicaffeoylquinic acid (CDCQ) mechanism on eNos and NO in endothelial cells.

**Figure 4 molecules-27-07954-f004:**
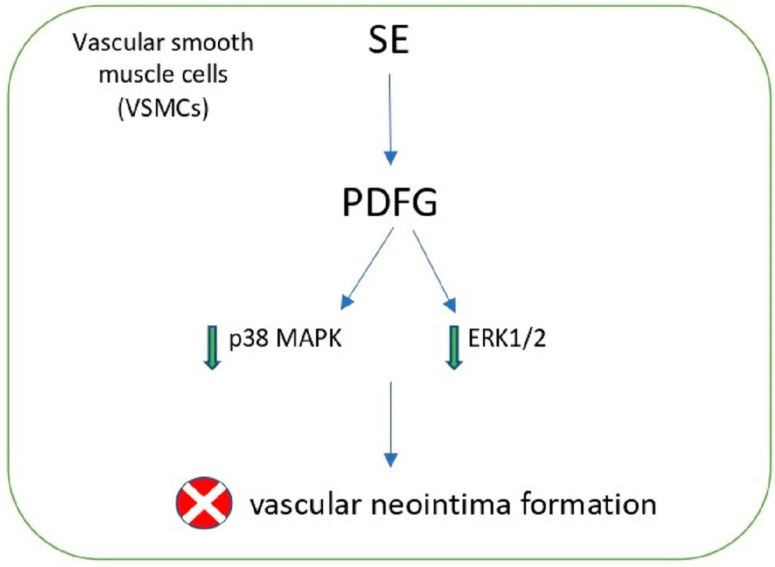
Neointima formation inhibition of Salicornia extract (SE) in VSMCs.

**Figure 5 molecules-27-07954-f005:**
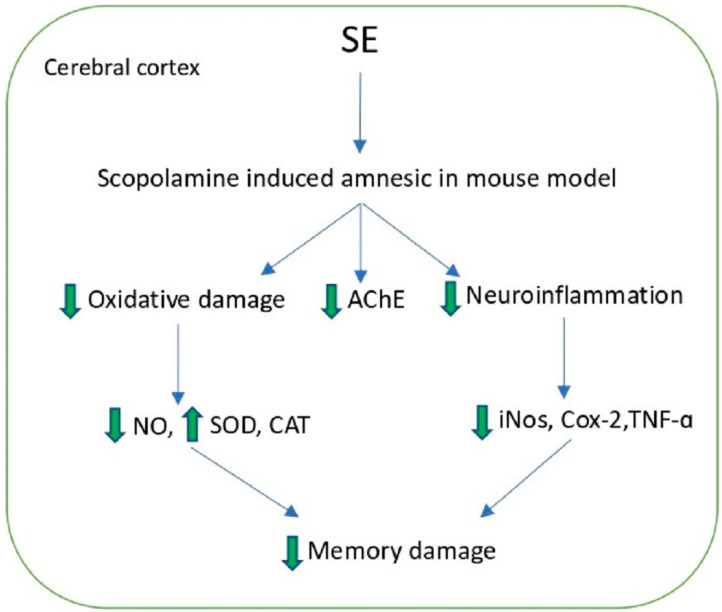
Ameliorative effect of Salicornia extract (SE) on Scopolamine induced amnesic.

**Table 2 molecules-27-07954-t002:** Extraction methods, quantities, and health-promoting effects of Salicornia polyphenols.

Polyphenolic Compounds	Salicornia Species	Extraction Method	Experimental Condition	Amount	Biological Activity	Ref.
Chlorogenic acid	*S. europea,* *S. ramosissima* *S. herbacea*	CE CE UAE CE MAE	EtOH H_2_O, 100 °C, 5 h 80% EtOH, 25 °C, 1 h H_2_O, 100 °C, 5 min 72–94 °C, 300 W, 5–10 min	0.22 mg/g 14.1 mg/g dw 53.19 ug/g fw 0.0758 mg/g dw 0.0342 mg/g dw	Antihypertensive Antimicrobic Reduction of neointimal hyperplasia	[24] [62] [90] [79]
Caffeoyl-5-dihydrocaffeoylquinic acid	*S. herbacea*	CE	MeOH, rt, 24 h	75.6 ± 2.3 mg/100 g fw	Inhibiting CE-OOH formation	[40]
3-caffeoyl-5-dihydrocaffeoylquinic acid methyl ester	*S. herbacea*	CE	MeOH, rt, 24 h	69 ± 1.4 μg/100 g fw	Inhibiting CE-OOH formation	[40]
3-caffeoyl-4-dihydrocaffeoylquinic acid methyl ester	*S. herbacea*	CE	MeOH, rt, 24 h	71.9 ± 1.9 μg/100 g fw	Inhibiting CE-OOH formation	[40]
3,5-dihydrocaffeoylquinic acid methyl ester	*S. herbacea*	CE	MeOH, rt, 24 h	171.9 ± 1.5 μg/100 g fw	Inhibiting CE-OOH formation	[40]
3-caffeoylquinic acid	*S. herbacea*	CE	MeOH, rt, 24 h		Inhibiting CE-OOH formation	[40]
3-caffeoylquinic acid methyl ester	*S. herbacea*	CE	MeOH, rt, 24 h		Inhibiting CE-OOH formation	[40]
3-caffeoyl-4-dicaffeoylquinic acid (tungtungmadic acid)	*S. herbacea*	CE	80% MeOH, rt	8 mg/kg dw	Protective effect on endothelial cell function	[43,63]
3,5-dicaffeoylquinic acid	*S. ramosissima*	UAE CE MAE	80% EtOH, 1 h, 25 °C H_2_O, 5 min,100 °C 300 W, 5–10 min, 72–9 °C,	25.83 mg/g fw 0.0259 mg/g dw 0.0280 mg/g dw	Antiproliferative Antihypertensive	[90] [79]
4,5-dicaffeoylquinic acid	*S. ramosissima*	UAE	80% EtOH, 1 h, 25 °C	11.75 mg/g fw	Antiproliferative Antihypertensive	[90]
*trans*-Ferulic acid	*S. europea,* *S. ramosissima,* *S. herbacea*	EAE CE UAE EAE + UAE CE CE MAE	H_2_O, 37 °C, 6 h H_2_O, 100 °C, 5 h 80% EtOH, 1 h, 25 °C 12 h, 40 °C and EtOH, 3 h MeOH, rt, 72 h H_2_O, 100 °C, 5 min 300 W, 72–94 °C, 5–10 min	2.60 ± 0.33 ug/g 8.2 mg/g dw 4.21 mg/g fw 8.45 mg% dw 0.1346 mg/g dw 0.0578 mg/g dw	Antidiabetic Antihypertensive Inhibitor of CYP450	[21] [62] [99] [63] [79]
*p*-Coumaric acid	*S. europea,* *S. ramosissima* *S. herbacea*	CE UAE CE CE MAE	H_2_O, 100 °C, 5 h 80% EtOH, 25 °C, 1 h MeOH, rt, 72 h H_2_O, 100 °C, 5 min 300 W, 72–94 °C, 5–10 min	3.19 mg/g dw 0.32 mg/g dw 2.75 mg/g fw 0.0483 mg/g dw 0.0349 mg/g dw	Antihypertensive Inhibitor of CYP450	[21] [62] [63] [79]
Pentadecylferulate	*S. herbacea*	CE	80% acetone, rt, 24 h.		Anticancer	[11]
Caffeic acid	*S. europea,* *S. ramosissima* *S. herbacea*	CE CE EAE + UAE CE MAE	EtOH H_2_O, 100 °C, 5 h 12 h, 40 °C + EtOH, 3 h H_2_O, 100 °C, 5 min 300 W, 72–94 °C, 5–10 min	0.28 mg/g dw 9.5 mg/g dw 6.87 mg% dw 0.0144 mg/g dw 0.0032 mg/g dw	Antibacterial Inhibitor of CYP450	[24] [62] [99] [79]
Gallic acid	*S. europea*	CE CE MAE	EtOH H_2_O, 100 °C, 5 min 300 W, 72–94 °C, 5–10 min	0.8 mg/g dw 0.21 mg/g dw 0.15 mg/g dw	Antibacterial Inhibitor of CYP450	[24] [79]
Protocathecuic acid	*S. europea,* *S. ramosissima* *S. herbacea*	CE EAE + UAE CE MAE	H_2_O, 100 °C, 5 h 12 h, 40 °C + EtOH, 3 h H_2_O, 100 °C, 5 min 300 W, 72–94 °C, 5–10 min	8.4 mg/g dw 1.54 mg% dw 0.1275 mg/g dw 0.0929 mg/g dw	Amelioration and prevention of vascular diseases.	[62] [99] [79]
Rutin hydrate	*S. europea,* *S. ramosissima*	CE CE MAE	EtOH H_2_O, 100 °C, 5 min 300 W, 72–94 °C, 5–10 min	10.05 mg/g dw 0.0999 mg/g dw 0.0781 mg/g dw	Antibacterial	[24] [79]
Catechin hydrate	*S. europea,* *S. ramosissima*	CE CE MAE	EtOH H_2_O, 100 °C, 5 min 300 W, 72–94 °C, 5–10 min	1.17 mg/g dw 0.1116 mg/g dw 0.0046 mg/g dw	Antibacterial	[24] [79]
Isoquercitrin 6”-O-methyloxalate	*S. herbacea*	CE	MeOH, rt, 24 h	0.47 mg/kg fw	Anti-inflammatory	[57]
Isorhamnetin 3-O-β-D-glucopyranoside	*S. herbacea*	CE CE	H_2_O, 100 °C, 5 h MeOH, rt, 24 h	16.2 mg/g dw 1.25 mg/kg fw	Anti-inflammatory Amelioration and prevention of vascular diseases.	[62] [57]
Quercetin 3-O-β-D-glucopyranoside	*S. herbacea*	CE CE	H_2_O, 100 °C, 5 h MeOH, rt, 24 h	3.4 mg/g dw 2.15 mg/kg fw	Amelioration and prevention of vascular diseases.	[62] [57]
Kaempferol	*S. europea,* *S.ramosissima* *S. herbacea*	UAE CE MAE CE	80% EtOH, 25 °C, 1 h, H_2_O, 100 °C, 5 min 300 W, 72–94 °C, 5–10 min MeOH, rt, 24 h	108.1–24.6 mg/100 g dw 10.90 mg/g 0.0052 mg/g dw 0.0047 mg/g dw	Inhibitor of CYP450	[90] [79] [63]
Quercetin	*S. europea,* *S.ramosissima* *S. herbacea*	CE EAE + UAE CE MAE	H_2_O, 100 °C, 5 h 12 h, 40 °C + EtOH, 3 h H_2_O, 100 °C, 5 min 300 W, 72–94 °C, 5–10 min	2.5 mg/g dw 12.63 mg%dw 0.0340 mg/g dw 0.0284 mg/g dw	Inhibitor of CYP450 Amelioration and prevention of vascular diseases.	[62] [99] [79]
Isorhamnetin	*S. europea* *S. herbacea*	CE EAE + UAE	H_2_O, 100 °C, 5 h 12 h, 40 °C + EtOH, 3 h	18.4 mg/g dw 6.65 mg% dw	Amelioration and prevention of vascular diseases.	[62] [99]
Acanthoside B	*S. europea*	EAE + CE	50 °C, 15 h + 50% EtOH, 3 h	2.40 mg/g	Neuroproective	[100]

CE: conventional extraction; UAE: Ultrasound assisetd extraction; MAE: Microwave assisted extraction; EAE: Enzyme assisted extraction; dw: dry weight; fw: fresh weight; rt: room temperature; CYP450: Cytochrome P450 System.

## Data Availability

Not applicable.

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
