# Peer review of "Overview of the Polyphenols in Salicornia: From Recovery to Health-Promoting Effect"

_molecules, 2022, doi:10.3390/molecules27227954_

Round 1

Reviewer 1 Report

The manuscript was prepared well; however, a few comments need to be addressed before acceptance, including

1- The mechanisms of action of most abundant polyphenol compounds against diseases are required by illustrating some mechanisms in Figures and text.

2- Try only to use the references of the last 10 years and delete the old ones (before 2011). 

Author Response

We would like to thank the reviewers for their useful comments and inputs that have surely improved the overall quality of our work. All the suggested issues have been faced and properly discussed in the revised manuscript. The corrections in the manuscript are shown in the “Tracked Changes” mode. Moreover, the detailed responses are listed below, point by point.

Reviewer 1

The manuscript was prepared well; however, a few comments need to be addressed before acceptance, including.

Thank you so much for your positive assessment. We have carefully considered your suggestions and edited the manuscript consequently.

The mechanisms of action of most abundant polyphenol compounds against diseases are required by illustrating some mechanisms in Figures and text.

Accordingly, in the revised version, we have illustrated three main mechanisms relating to the action of phenolic acids (Figure 3, lines 216-221), flavonoids (Figure 4, lines 336-346), and lignans (Figure 5, lines 394-400), respectively.

Try only to use the references of the last 10 years and delete the old ones (before 2011).

Actually, we have tried to refer to the most recent articles dealing with Salicornia. We have deleted the references older than 2011, except for a few cases in which no recent works are available.  We mean the refs 55, 91, and 99.

Reviewer 2 Report

1-It would be nice if it was clearly stated in the title that there is a compilation.

2-The title of Chapter 2 after the introduction is too general, wouldn't the title of 'principal phytochemicals (especially polyphenols) identified in Salicornia' be more appropriate instead? Line 69

3- First of all, information about Salicornia should be given. How many species are there? What are their differences? What properties? It can be even more memorable with a photo. After these are given, it would be more appropriate if the studies done by entering the 2nd Chapter were given.

4-Line 459 enzyme name is misspelled. The enzyme name should have been hemicellulase.

5-Table 1 is not referenced.

6-Tables and figures should be given after the subject mentioned. These should be cited before tables and figures.

7- Since it is a molecule magazine, it would be nice if the chemical structures of the specified components were also given. At least the shape and molecular structures of some unknown phenolic acids could be given.

8-What is the situation in terms of oxalate and saponin? What is the effect on health? What should the dosage be?

Author Response

We would like to thank the reviewers for their useful comments and inputs that have surely improved the overall quality of our work. All the suggested issues have been faced and properly discussed in the revised manuscript. The corrections in the manuscript are shown in the “Tracked Changes” mode. Moreover, the detailed responses are listed below, point by point.

Reviewer 2.

It would be nice if it was clearly stated in the title that there is a compilation.

According to your suggestion, we have modified the title as such: “Overview of the polyphenols in Salicornia: from recovery to health-promoting effect”.

The title of Chapter 2 after the introduction is too general, wouldn't the title of 'principal phytochemicals (especially polyphenols) identified in Salicornia' be more appropriate instead? Line 69.

Agree with you. We have modified the subtitle as suggested.

First of all, information about Salicornia should be given. How many species are there? What are their differences? What properties? It can be even more memorable with a photo. After these are given, it would be more appropriate if the studies done by entering the 2nd Chapter were given.

Accordingly, we have summarized the main characteristics of the most studied Salicornia species in Table 1 (with the help of photos) and in the text (lines 34-45) of the revised manuscript.

Line 459 enzyme name is misspelled. The enzyme name should have been hemicellulase.

You are right, we have corrected the enzyme name in the revised version (line 590).

Table 1 is not referenced.

We are sorry for the mistake. The new Table 2 was correctly referenced in the revised text (lines 601-603).

Tables and figures should be given after the subject mentioned. These should be cited before tables and figures.

We have revised the text and carefully checked the order and position of figures and tables.

Since it is a molecule magazine, it would be nice if the chemical structures of the specified components were also given. At least the shape and molecular structures of some unknown phenolic acids could be given.

Surely, we agree with this observation, indeed we grouped together the structures of the main polyphenols (including some uncommon phenolic acids, such as tungtungmadic acid) in Figure 2.

What is the situation in terms of oxalate and saponin? What is the effect on health? What should the dosage be?

Regarding this request, we have added some information in lines 156-161 of the revised manuscript.

Round 2

Reviewer 1 Report

accept